# Life Cycle Cost and Assessment of Alternative Railway Sleeper Materials

Samuel Thompson [1], Christopher King [1], John Rodwell [2], Scott Rayburg [1,*] and Melissa Neave [3]

[1]  Department of Civil and Construction Engineering, Swinburne University of Technology,
     Hawthorn 3122, Australia; 102304539@student.swin.edu.au (S.T.); 101107575@student.swin.edu.au (C.K.)
[2]  Department of Management & Marketing, Swinburne University of Technology, Hawthorn 3122, Australia;
     jrodwell@swin.edu.au
[3]  School of Global, Urban and Social Studies, RMIT University, Melbourne 3001, Australia;
     melissa.neave@rmit.edu.au
*   Correspondence: srayburg@swin.edu.au; Tel.: +61-(3)-9214-4944

**Abstract:** Improvements in plastic recycling technology along with pressure to reduce emissions and waste has led to a desire to find environmentally friendly, cost competitive railway sleepers. This study conducts life cycle analyses of emissions and costs for timber, concrete, short fibre and long fibre composite railway sleepers to determine which sleepers are more environmentally friendly and cost competitive. The results clearly highlight the environmental advantages of short fibre plastic composites. The basic scenario had concrete sleepers being the most cost competitive, before factoring in the recyclability and likely future cost reductions of short fibre composite sleepers. With as little as 50% of the entirely recyclable short fibre sleepers being recycled their cost quickly becomes comparable to concrete sleepers. Further, there are several likely changes in the future that will make short fibre sleepers even more cost competitive. The short fibre industry is still growing and could substantially reduce costs through the effects of economies of scale and experience curves of production. A further driver of future cost competitiveness would be the broader use of an Australian or international carbon price, where concrete sleepers have a disadvantage. Together, these changes indicate that short fibre composites have great potential financially and environmentally.

**Keywords:** short fibre plastics; plastic recycling; concrete sleepers

## 1. Introduction

Of the many environmental issues facing humanity, two of the more concerning are climate change and plastic pollution. These two environmental challenges have far reaching consequences for human and ecosystem health, and solutions to these problems will need to come from across the spectrum of human endeavour [1]. One way to help resolve both of these issues is to use new products that have lower embodied emissions while also finding new ways to reuse plastic waste. The recycling and reuse of plastic is urgent as, in 2018 alone 359 million tonnes of plastics were produced, an increase of 3% on the previous year [2]. If plastic is not recycled and reused, it accumulates in local and global ecosystems causing harm to natural habitats. Given that transport is one of the largest contributors to global greenhouse gas emissions [3], one way to address plastic pollution would be to find new ways to use reclaimed plastic that improve transport networks while also reducing greenhouse gas emissions [4].

To help reduce transport emissions, the increasing use of rail transport is a necessity because it is one of the lowest emission forms of transport [5]. However, rail is not completely emissions free, with one of the largest sources of rail emissions coming from rail infrastructure, especially the use of railway sleepers [6]. Traditionally, sleepers were produced with treated hardwood timber, with timber sleepers first used for rail infrastructure in the 1800's. However, due to a scarcity of wood around the 1880's, steel sleepers

became more common, although timber sleepers continued to be deployed worldwide [7]. Although at first glance timber sleepers might seem to be an environmentally friendly option, the harvesting and processing of the timber generates substantial waste. Moreover, because timber sleepers are chemically treated prior to installation when they reach their end-of-life stage it is often not possible to reuse them and they are difficult to discard [8]. These reusing and recycling challenges, coupled with the short life span of timber sleepers, which last only ten to twenty years, means that timber sleepers can have a large negative environmental impact [9].

One of the most common timber sleeper replacements in use today are concrete sleepers. Concrete sleepers were introduced in Britain, France, and Germany as early as 1943 due to the scarcity of timber during World War II, but prestressed concrete sleepers were only adopted in the United States in 1966 [10]. Concrete sleepers have since become one of the most common types of railway sleepers installed worldwide owing to their increased life span and strong structural performance. However, the production of concrete sleepers may release in the range of 10 to 200 times more carbon dioxide equivalent ($CO_2e$) than treated hardwood sleepers [11,12]. Hence, while concrete sleepers offer some conveniences over timber sleepers, they are exacerbating the climate crisis through their high carbon emissions and, as such, they reduce the environmental benefits of rail as a transport option.

A possible solution to this problem, as well as that of plastic pollution, is the development of composite sleepers assembled from recycled waste plastics [13]. Recycled composite materials have been used for the development of railway sleepers in Japan since the 1980's [14], although the use of this material has not been considered a realistic alternative in many countries due to perceived limitations of the material [7]. However, technological advancements coupled with decades of research have resulted in improved products that mitigate many of those limitations, making this option potentially more appealing to railway industries.

The global market for composite sleepers is currently expanding at a rapid pace as research into these technologies created opportunities to improve upon the limitations of traditional sleepers, yet in Australia there are still few suppliers of these products due to a lack of demand. The lack of uptake in Australia is causing limited production, which in turn can cause the unit price of these products to be higher than traditional sleepers. However, at this stage, no study has investigated the relative emissions benefits of a shift to composite sleepers, nor whether such sleepers are cost competitive in Australia. Thus, it is unclear where the products might fit into the Australian market and whether they represent a viable method for reducing transport emissions and the generation of plastic pollution.

The purpose of this study is to address this problem by assessing the life cycle emissions and costs of composite and traditional sleeper technologies and determining whether a shift to composite sleepers is warranted and financially viable. To achieve these goals, the specific objectives of this study are to:

(1) conduct a Life Cycle Assessment (LCA) on the emissions of traditional and composite sleepers,
(2) conduct cost analyses of traditional and composite sleepers that considers the purchase, installation and life span of these products, and
(3) determine whether composite sleepers are more environmentally friendly than traditional sleepers and whether they can be cost competitive.

## 2. Materials and Methods

The methods used to assess traditional (timber and concrete) and composite (short and long fibre) sleepers in this study are life cycle emissions assessment and life cycle cost assessment. This section explains these methods in more detail and includes the equations used to calculate the emissions and costs over the design life of the products.

This study considers two types of composite sleepers, namely short and long fibre reinforced sleepers. Short fibre composites are sleepers with fibre reinforcement of less than twenty millimetres. Within this study the short fibre composite material contained no reinforcement at all, however it still falls under the naming convention of short fibre [15]. The short fibre composites considered in this study are produced within Victoria, Australia. These composite sleepers contain approximately 85% rigid and flexible recycled plastic materials with the remaining contribution coming from virgin resources [16]. This product contains no fibre or steel reinforcement meaning that the sleepers can easily be fully recyclable at the end of their design life, although due to the length of the current life there has been no opportunity to do implement this as yet [16]. The absence of reinforcement means that these products fall short when compared to concrete and timber sleepers in terms of strength, stiffness, and dynamic properties [7] and these shortcomings are a factor that currently limits the application of these sleepers. However, progress on assessing whether these limitations still exist is being made with track testing occurring in freight lines in the Toowoomba region, at Richmond station, and at a mainline stabling facility at Wyndham Vale [16]. The culmination of this testing subsequently resulted in type approval for mainline rail operators V/Line and Metro Trains Melbourne in 2021.

The long fibre composite sleepers considered in this study are made from fibre-reinforced foamed urethane (FFU). These sleepers are produced by coating continuous longitudinal glass fibre in polyurethane and combining them to form a sleeper. The glass fibres add strength that is lacking in the short fibre sleepers, which makes the mechanical and physical properties of the long fibre sleepers more similar to those of timber sleepers. The glass fibres do, however, have a considerable environmental impact [15]. Long fibre FFU sleepers were originally developed and first implemented in Japan's railway network in the 1980's and they are now used throughout that country, including for highspeed rail lines. These types of sleepers are also now found throughout the world but are often less utilised than timber or concrete sleepers due the considerably greater initial purchase price and the difficulties associated with the handing of fibreglass. There are currently no manufacturers of FFU sleepers in Australia and therefore these products must be imported from Japan, which further increases their environmental and economic costs. These production issues lead to the price of these products being up to ten times more than a typical timber sleeper, which greatly hinders their mass application [7].

### 2.1. Life Cycle Assessment

An LCA will be used to assess the emissions of short and long fibre composite sleepers, and timber and concrete sleepers. The use of an LCA will bring a more inclusive and quantitative method of assessment to the environmental impacts of these products [17]. Each product will be considered across its whole life cycle, including the raw material acquisition, manufacturing, implementation and use throughout their operational life, and the removal processes that occurs when the product reaches the end of its effective life [18]. By segmenting the process across its life cycle, specific outlying contributors will be efficiently identified while also allowing for the cumulative environmental impact to be tabulated [19]. As previously mentioned, these LCAs will focus on the greenhouse gas emissions of each sleeper. There are several different greenhouse gases (GHGs) produced over a sleeper's life cycle and these each have different impacts on global warming [20]. To accommodate this variability, emissions will be reported as $CO_2$e (or $CO_2$ equivalent). Carbon dioxide equivalency allows emissions from GHGs that are not carbon dioxide, to be equated based on each gas's global warming potential. $CO_2$ equivalent is the functional unit of the LCAs and is recorded in kilograms of carbon emissions produced.

The LCA process adopted in this study will follow the form set out in AS ISO 14040:2019, Environmental management–Life cycle assessment–Principles and framework. However, due to the conceptual and simplified nature of the assessment that will be undertaken (given the sometimes-limited availability of data), these results are not expected to qualify for AS ISO 14040:2019. However, the LCAs developed in this study will include

all stages from the cradle to grave, and back to the cradle wherever possible, as shown in Figure 1.

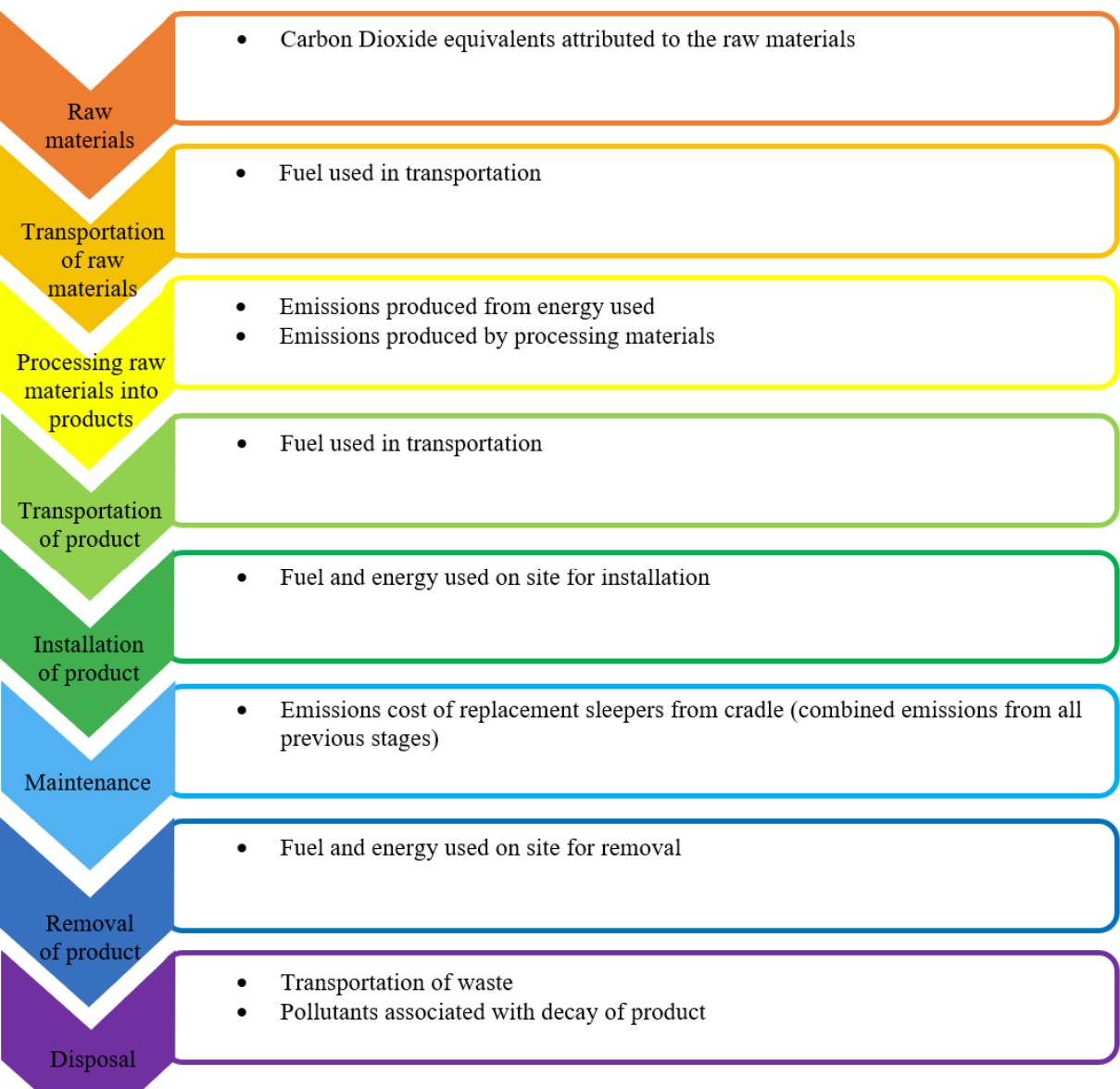

**Figure 1.** The steps included in the Life Cycle Assessment performed in this study.

Although some LCAs have been previously undertaken for traditional sleepers [21,22], new LCAs were generated in this study to ensure that the LCAs for all of the sleepers investigated are directly comparable and use the same method of calculation. For this assessment, the dimensions of both the timber and concrete sleepers fall within the bounds of relevant Australian and International Standards. Thus, the dimensions of the timber sleepers match those specified in the RailCorp engineering specifications for timber sleepers and bearers document (SPC 231) and are provided in Table 1. These timber sleepers have been designed in accordance with AS 3818.1 and AS 8313.2 and are the standard for the state of New South Wales, Australia.

**Table 1.** Minimum dimensions and tolerances for sleepers SPC 231.

| Standard Gauge Hardwood Sleeper | |
|---|---|
| Length | 2440 mm |
| Width at Base | 230 mm |
| Depth at Centre of Rail Seat | 130 mm |
| Approximate Weight | 65 kg |

The dimensions that have been chosen for the control prestressed concrete tie-downs are from John Holland Group's engineering specifications for concrete sleepers document (CRN CN 232) and these data are provided in Table 2. These specifications have been selected as they also meet the Australian Standards AS 1085.14. To equitably compare the traditional and composite sleepers the dimensions for both types of composite sleepers match the timber sleepers presented in Table 1.

**Table 2.** Standard sleeper dimensions CRN 232.

| Medium Duty Concrete Sleeper | |
|---|---|
| Length | 2390–2500 mm |
| Width at Base | 220–255 mm |
| Depth at Centre of Rail Seat | 180 mm |
| Approximate Weight | 285 kg |

### 2.1.1. Sleeper Materials

The embodied emissions of the sleeper materials investigated in this study are calculated using Equation (1):

$$E_1 = \Sigma \, M_i \times f_i \tag{1}$$

where $E_1$ is the total embodied greenhouse gases in carbon dioxide equivalents (kgCO$_2$e), Mi is the mass or volume of each material in units of either kilograms or cubic meters, and fi is the carbon dioxide emissions factor (kgCO$_2$e/unit). Values of fi are derived from the Australian Life-Cycle Index. All emissions factors from the Life-Cycle Index include only the emissions that the materials embody prior to product transportation.

For composite sleeper materials, when the sleepers reach the end of their useful life the materials themselves may be recycled for use in the production of new sleepers. This study will consider how the recyclable nature of composite sleepers may impact their LCA emissions by considering a scenario where 50% of composite sleeper materials are reclaimed. Further, the material used in timber sleepers can also be recycled and used to produce more sleepers or the sleepers may also be repurposed for domestic uses, such as in garden beds (depending on the pre-treatment processes used to prepare them and their post-use condition). To accommodate this potential second life, a scenario where the timber is reclaimed at a rate of 50% is also be considered.

### 2.1.2. Sleeper Material Processing

Where not already included in the results for the materials themselves, the processing of raw materials is also assessed in these LCAs. Discretion will be used to determine when an appropriate result or emissions factor has already been established that includes processing. For the raw material cement, which is used in the construction of concrete sleepers, previously completed LCAs have established an emissions factor that includes all of the embodied emissions from the input materials and material processing. Hence, in this case, cement can be taken as a raw material rather than needing to individually calculate the emissions of the input materials: limestone, clay and shale; and the processes that result in the final product. In all other cases, Equation (2) was used to determine the material processing emissions. Equation (2) is used to calculate the carbon dioxide

equivalents for the electricity used in the manufacturing process to turn the raw material into a finished product:

$$E_2 = \Sigma \, M_j \times f \tag{2}$$

where $E_2$ is the total emissions from manufacturing ($CO_2e$), $M_j$ is the electricity that is purchased from the provider and factors in the electricity that is lost in transmission to the manufacturing plant (kWh), and f is a factor that adjusts for the region in which the plant is located, and which is based on the type of energy used in the region. For example, for green energy such as solar, f would be 0, while in Victoria, Australia, where brown coal is typically used to produce electricity, the factor is 1.13 ($kgCO_2e$/kWh).

### 2.1.3. Fastening Systems

Sleeper fastenings or fixings are a vital component of railway infrastructure that holds the rail tracks to the sleepers. When calculating the emissions of the sleepers it is also important to consider their fastenings because no type of sleeper can be used without them. Moreover, fastenings come in different forms and thus have different environmental impacts. Concrete sleeper fastenings are embedded during the casting of the sleepers while timber and composite sleepers have the fastening systems inserted on site during the installation phase. To account for the impact of the fastening system, Equations (1) and (2) will be used again but this time considering the materials used for the construction of the fastening systems components. For this study, it will be assumed that each concrete sleeper uses the Pandrol E Clip cast as part of the fastening system [23]. For all other sleepers: two rail clips, two baseplates, and four screw spikes will be assumed to be used per sleeper. Fastening systems have a greater design life than the sleepers themselves, therefore this study assumes that when the sleepers reach the end of their life some of the fastening systems can be reused. Consequently, the study considers two cases: one where no fastenings are reused and another where 50% of fastenings are reused.

### 2.1.4. Transportation

Transportation takes place during many stages of the product life cycle of sleepers. To account for these emissions, Equation (3) will be used:

$$E_3 = \Sigma \left[ \left( \frac{D_m}{FE_m} \right) \times \left( \frac{M_m}{C_{cap}} \right) \right] * f_{trans} \tag{3}$$

where $E_3$ is the total transport emissions ($CO_2e$), $D_m$ is the total distance travelled from initial loading to installation point (km), $FE_m$ denotes the fuel efficiency of the mode of transport (km/l), $M_m$ is the amount of material being transported (kg), $C_{Cap}$ represents the capacity of each mode of transport with a defined number of units per load as defined by standard industry practice, and $f_{trans}$ is the emissions factor for the vehicle used for transposition ($kgCO_2e$/tkm). For the purposes of this study, that transport will be by truck with an average distance travelled of 100 km from place of production to point of installation.

### 2.1.5. Installation and Removal

Equation (4) represents the emissions produced in $kgCO_2e$ by the installation and removal phase of life:

$$E_4 = E_{operation} + E_{idle} = \Sigma (L_{operation} \times f_i) + (L_{idle} \times f_i) \tag{4}$$

where $E_{operation}$ is the emissions produced by the operation of the machinery (kgCO2e) and $E_{idle}$ is the emissions produced by the machinery at idle (kgCO2e). The fuel used for operation, $L_{operation}$, and idle, $L_{idle}$ (l), is then multiplied by the carbon dioxide emissions factor fi (kgCO2e/l). Due to the theoretical nature of this study, fuel used during these activities was not measured but rather fuel consumption threshold figures were taken from equipment manufacturers. For the purposes of this study, the installation of sleepers is

assumed to take place on an existing rail line where there are already sleepers in place that require removal. To perform the removal and subsequent installation it is expected that the use of a wheeled excavator and frontend loader will be required.

2.1.6. Final Emissions Calculations

Once Equations (1)–(4) have been used to find the E1, E2, E3, and E4 for an individual sleeper of each type of material, Equation (7) will be used to combine the results to get the total embodied emissions for an individual sleeper, $E_T$:

$$E_T = E_1 + E_2 + E_3 + E_4 \tag{5}$$

ET will provide an indication of the relative environmental impact of an individual sleeper. However, to gain a more realistic result applicable to a real-world infrastructure scenario, the results of ET need to be modified. Due to the differing physical characteristics and capacities of the different sleeper types, as specified by the Australian Standards, the sleepers are required to be placed at different spacings along the railway line. The spacing of sleepers can also vary depending on the location and use. To reduce this variability, this study assumes that the sleepers are placed for mainline use with maximum speeds of 80 km/h or maximum axle loads of 25 tonnes. Within Victoria, this requires timber sleepers to be placed at a distance of 0.685 metres apart and concrete sleepers to be placed at 0.72 metres centre to centre [24]. The Australian Rail Track Corporation does not specify the spacing required for composite sleepers, however, their comparable densities and physical performance suggests that they should be placed at a distance similar to that of timber. To standardise for these differences, this study will compute emissions per 100 m of track, rather than per sleeper. Over this length, 146 timber or composite sleepers are assessed compared to 139 concrete sleepers.

Finally, each sleeper has a different expected design life as specified by the manufacturer. These different design lives should also be factored into each individual $E_T$ value by multiplying it by the number of times the sleepers require replacement, plus the initial installation, over the project's life span. The project life span for this study will be set at 100 years.

*2.2. Life Cycle Cost Assessment*

In addition to assessing the emissions of different sleeper types, this study also seeks to determine the relative cost of implementing each sleeper technology. To this end, the study adopts a Life Cycle Cost Assessment (LCCA) approach. An LCCA is a method that assesses the total cost of owning a facility or the cost of a project. In this instance, the project considered is a 100 m length of track and an LCCA will be used to evaluate the total cost of each sleeper variant that could be used for that section of track. The LCCAs will include the initial cost, any related service costs, preventative maintenance costs, operating costs, and the cost of disposal at the end of use. LCCAs are used for instances in which there are numerous alternatives for a project to reach an outcome, yet each alternative has different initial and ongoing costs. A major component of LCCA is value engineering that aims to outline the project's cost to significantly lower expenditure. This method is used as it outputs data that is comprehensive and useable. An example of an LCCA used for a purpose similar to that of this study is the case by [25].

2.2.1. Sleepers

For each type of sleeper, the initial cost of the sleeper will be set as the purchase price. The costs associated with sleepers include the cost of production, the wholesale price and the retail price, but for a railway system the initial purchase price will typically be the retail price. Although there is existing literature containing the prices of both hardwood timber and concrete sleepers, to gain a more current representation of the retail price suppliers were contacted and prices were retrieved via personal communications. No literature was found containing the price of either composite sleeper type, thus again the method of

personal communication with suppliers was required to obtain prices. The obtained price information did require some modification to be in a usable form. For long fibre composites prices were supplied in a cubic meter rate that was converted based on the dimensions in Table 1. Short fibre composites provided greater challenges as the price of a standard gauge sleeper is not publicly available. To overcome this the known price of a narrow gauge sleeper of the same material was scaled up based on the increase in volume from narrow to standard gauge. The initial purchase price for the sleeper is represented as C1 within the calculations.

### 2.2.2. Fastening System

Each railway sleeper uses two rail clips, two baseplates and four screw spikes. These costs are assumed to be the same for timber and both types of composite sleepers. The accrued costing for these components shall be taken as $C_2$. Meanwhile, concrete sleepers do not accrue a cost for fastenings as they are provided with the purchase of the sleepers and as such the fastening costs are included in the retail price of those sleepers.

### 2.2.3. Transportation

Transportation of the sleepers only accounts for the cost of the transportation of the sleepers to the site of installation. The purchase of company vehicles and the depreciation of these vehicles has been excluded as they are considered costs to the business. Therefore, the only costs that are included in this study are the cost of transit, which can be either the cost of the fuel consumed or the cost of third-party delivery. For all sleepers, a standardised distance travelled of 100 km was used in this calculation, meaning that transport price differences are instead based on the size and weight of the sleepers relative to each other. The fuel used by cargo carrying modes of transport is predominantly diesel. Consequently, this analysis used an average price of diesel in Melbourne Victoria over the month of June 2021, which was 153 cents per litre. The cost of third-party delivery is included in the price $C_1$ whereas $C_3$ denotes the cost of transport. The importation cost of the long fibre composite sleepers from Japan is included in the cost of the cost of purchase, $C_1$.

### 2.2.4. Installation and Removal

The installation and removal of sleepers have been combined into a single cost because the installation or replacement of sleepers automatically requires the removal of sleepers that are already present and at their end of life. The cost of installation and removal does not consider the purchase of the equipment that removes and installs the sleepers (which is already owned) and therefore only the cost of the fuel (diesel) to run the machines required to perform the work is assessed. The cost of labour associated with the removal and installation has also been considered. While dependant on the company tasked with installing the sleepers the cost of labour can vary. However, it is expected that approximately the same amount of labour will be required to perform the installation and removal per sleeper regardless of the type. As such, while the accuracy to all real world labour costs may not be perfect, the precision within the study is reliable. The cost of installation and removal is denoted by $C_4$.

### 2.2.5. Final Cost Calculation

As previously mentioned, this study assumes a project lifespan of 100 years. The total cost of replacement for each sleeper over this time period is $C_T$ and is calculated using Equation (6). The equation only pertains to the design life of each sleeper type. The change in design life between each sleeper type will determine the total cost of the project for the use of that type of sleeper. The inflation rate is assumed to 2% per annum and is also accounted for when calculating the current life cycle for the 100 year project life.

$$C_T = C_1 + C_2 + C_3 + C_4 \tag{6}$$

The project considers a length of track that spans 100 metres and that is already installed. The equipment used to remove sleepers at the end of life and install new sleepers differs depending on the application. For longer spans of track or track that needs ballasts and sleepers and only has the railway track laid in place, such as the construction of new railway lines, more specialised equipment is used.

## 3. Results

This study includes two types of LCAs, an emissions focussed LCA and a cost focussed LCA. This section presents the results of these two analyses in turn.

### 3.1. Life Cycle Assessment

The emissions focussed LCA was computed using Equations (1) to (5). The analysis also included a consideration of individual product life spans (which controls the number of installations required over a 100-year timeframe) and the number of sleepers required per 100 m of track as specified by rail authority sleeper spacing. These details are outlined in Table 3. Table 4 and Figure 2, present the total embodied emissions for each sleeper type using this approach.

**Table 3.** Design life of railway sleeper variables.

| Sleeper Type | Number of Sleepers | Design Life Span (years) | Required Installations per 100 years (#) |
|---|---|---|---|
| Hardwood | 146 | 20 | 5 |
| Concrete | 139 | 50 | 2 |
| Short Fibre Composite | 146 | 50 | 2 |
| Long Fibre Composite | 146 | 50 | 2 |

These data show that sleeper manufacturing is the largest source of emissions for all sleeper types and that long fibre composites have the largest manufacturing emissions, followed by hardwood, with short fibre composites having the lowest manufacturing emissions. Fastening emissions were similar across all types of sleepers with the exception of hardwood sleepers, whose fastening emissions are roughly twice those of other sleeper types.

**Table 4.** Embodied emissions for sleepers under design case.

| Sleeper Type | Sleeper ($tCO_2e$) | Fastening ($tCO_2e$) | Transport ($tCO_2e$) | Install/Removal ($tCO_2e$) | Total ($tCO_2e$) |
|---|---|---|---|---|---|
| Hardwood | 40.32 | 15.98 | 1.47 | 15.25 | 73.02 |
| Concrete | 23.61 | 6.84 | 7.30 | 5.81 | 43.55 |
| Short Fibre Composite | 11.68 | 6.39 | 0.68 | 6.10 | 24.85 |
| Long Fibre Composite | 163.16 | 6.39 | 2.89 | 6.10 | 178.54 |

Transport emissions were much higher for concrete sleepers than other types of sleepers owing to their weight with the lowest transport emissions found for short fibre composite sleepers. Installation and removal emissions were again, roughly uniform across sleeper types with the exception of hardwood sleepers, whose emissions for installation and removal were much higher. Overall, total emissions were the highest for long fibre composite sleepers, followed by hardwood and concrete sleepers with the lowest emissions found for short fibre composite sleepers. If we consider concrete as the current default sleeper type, the emissions for long fibre composites are 4x higher, the emissions for hardwood are 1.7x higher and the emissions for short fibre composites are 40% lower.

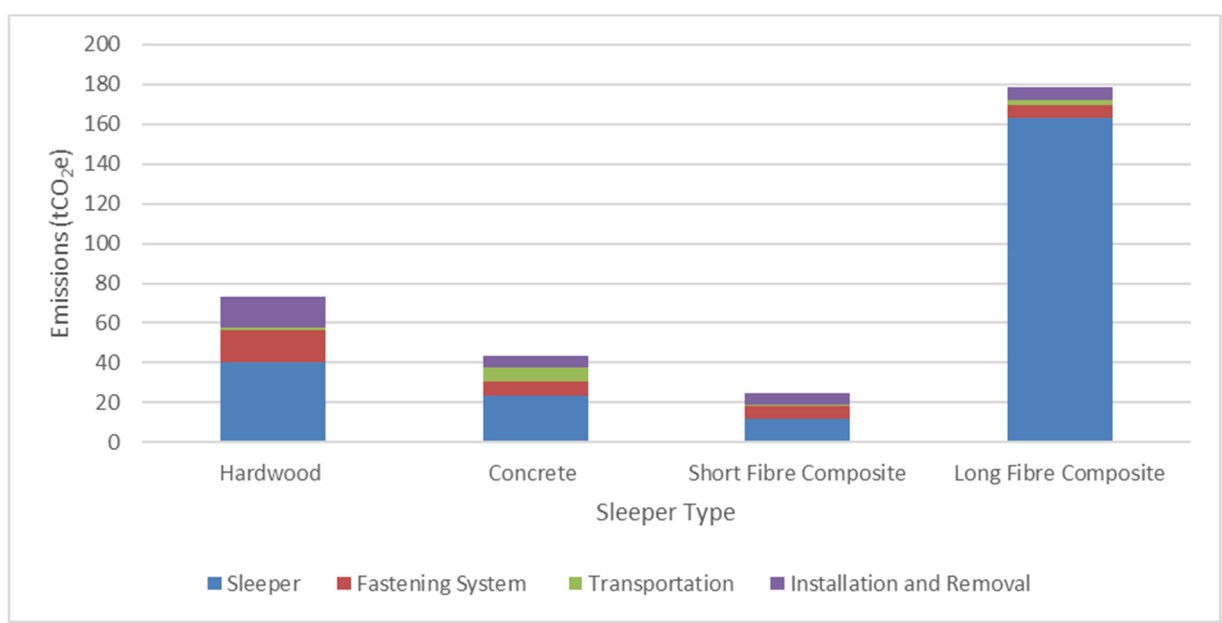

**Figure 2.** Total embodied emissions for each sleeper type under design case.

Next, the emissions were recalculated for each sleeper type while assuming that 50% of fastening systems could be reclaimed, 50% of the materials used in creating composite sleepers could be used to create more composite sleepers, and 50% of timber sleepers could be reclaimed for domestic uses (Figure 3, Table 5).

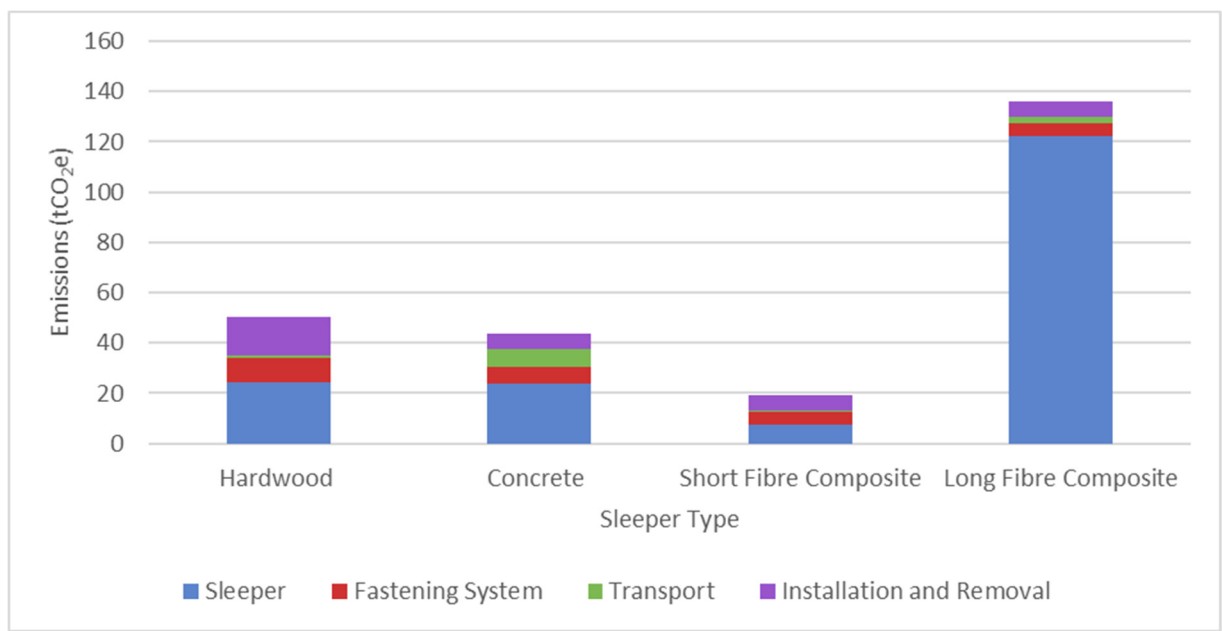

**Figure 3.** Total embodied emissions for sleepers with 50% fastening reuse and material reclaiming.

**Table 5.** Embodied emissions for sleepers with 50% fastening reuse and material reclaiming. This rate of reuse results in reduced $tCO_2e$ of 60%, 65% and 75% of relative to the non-reuse scenario for Hardwood, Short Fibre- and Long Fibre Composites respectively.

| Sleeper Type | Sleeper ($tCO_2e$) | Fastening ($tCO_2e$) | Transport ($tCO_2e$) | Install/Removal ($tCO_2e$) | Total ($tCO_2e$) |
|---|---|---|---|---|---|
| Hardwood | 24.19 | 9.59 | 1.47 | 15.25 | 50.50 |
| Concrete | 23.61 | 6.84 | 7.30 | 5.81 | 43.55 |
| Short Fibre Composite | 7.59 | 4.79 | 0.68 | 6.10 | 19.17 |
| Long Fibre Composite | 122.37 | 4.79 | 2.89 | 6.10 | 136.16 |

These results show that the pattern remains the same (with the most emissions intensive sleeper remaining long fibre composite and the least emissions intensive sleeper being short fibre composite), but the gaps between sleeper types have changed. Now, short fibre composite sleeper emissions are about 45% of those of concrete sleepers while the difference in emissions between timber and concrete sleepers has narrowed with hardwood sleeper emissions being only 14% higher than concrete sleepers.

*3.2. Life Cycle Cost Assessment*

Using the outlined method, the life cycle costs for each sleeper have been calculated. In line with the life cycle emissions results, the same sleeper requirements (Table 3) were used when calculating life cycle costs. The results of this analysis are presented in Table 6. As the project life span for this study was set to be 100 years, it was necessary to adjust for inflation. To this end, it was assumed that inflation will be 2% per annum for the entire 100-year period although in reality this will vary depending on the economic conditions that prevail over this time frame.

**Table 6.** Life cycle-cost for the base sleeper design case.

| Sleeper Type | Sleeper ($AUD) | Fastening ($AUD) | Transport ($AUD) | Install/Removal ($AUD) | Total ($AUD) |
|---|---|---|---|---|---|
| Hardwood | 168,855.01 | 37,523.33 | 2127.33 | 228,424.05 | 436,929.72 |
| Concrete | 71,838.30 | - | 7576.26 | 62,473.87 | 141,888.43 |
| Short Fibre Composite | 109,411.26 | 10,779.43 | 707.36 | 65,620.03 | 186,518.12 |
| Long Fibre Composite | 350,331.70 | 10,779.43 | 502.48 | 65,620.03 | 427,233.66 |

These results show that the concrete sleepers are the most cost effective over their life cycle followed by short fibre composites, with timber and long fibre composites coming in at about the same total cost. Short fibre sleepers were found to be about 30% more expensive over their life cycles than concrete sleepers whereas timber and long fibre composite sleeper costs are more than triple the cost of concrete sleepers. The main reason why concrete sleepers were the cheapest is their initial purchase price, which was the lowest of all sleeper types and already includes fasteners. Whilst the transport phase for concrete sleepers was much higher than that for other sleepers, this was offset by the fastening systems being included in their purchase price. Meanwhile, installation costs are between 3–4 times higher for timber sleepers (owing to their short lifespans) than all other sleeper types, the remainder of which have very similar installation costs.

When finding purchase prices, some were available in the literature although personal communication with manufactures, suppliers, and rail construction companies was required to obtain all figures. The values presented here are an accurate representation of sleeper prices at the time of publication of this study although it is acknowledged that there are unique variables and conditions that impact the purchase price of products. As previously mentioned, concrete sleeper fastenings are cast into the sleepers themselves, so the purchase price includes the fastenings. Alternatively, fastenings must be purchased alongside the sleepers for both composite types and for hardwood timber sleepers. While

these three types of sleepers are assumed to use the same fastenings, replacement frequency does impact the project life span cost. To be consistent with the LCA emissions calculations for fastenings, it is assumed that virgin fastenings are used and thus must be purchased every time a sleeper is replaced. Transportations costs equate to the cost of diesel fuel required for the trucks used to transport the sleepers and not the purchase price of the trucks themselves. Long fibre composites require freight shipping to Australia however this cost is already included in the purchase price. Labour costs for truck operators have not been considered within the calculations for transport costs. Installation and removal costs are based on the cost of fuel used to operate the necessary machinery and the cost of the associated labour. Labour costs includes the use of labourers, machine operators, supervisors, and safety coordinators.

Next, as with the LCA for sleeper emissions, the calculations were redone assuming that 50% of fastening systems were reused and materials were reclaimed at the same rate for composite and timber sleepers. The results of this analysis are presented in Table 7. These data show that short fibre composite sleeper prices are now just above the cost of concrete sleepers. Meanwhile, the costs for timber and long fibre composite sleepers have decreased, but they remain much more expensive than the other two options. These data demonstrate that the reclamation and reuse of materials can have a significant impact on life cycle costs for sleepers.

**Table 7.** Life cycle-cost for sleeper design case adjusted for 50% fastening reuse and material reclaiming.

| Sleeper Type | Sleeper ($AUD) | Fastening ($AUD) | Transport ($AUD) | Install/Removal ($AUD) | Total ($AUD) |
|---|---|---|---|---|---|
| Hardwood | 168,855.01 | 20,221.67 | 2127.33 | 228,424.05 | 419,628.06 |
| Concrete | 71,838.30 | - | 7576.26 | 62,473.87 | 141,888.44 |
| Short Fibre Composite | 69,524.64 | 6849.72 | 707.36 | 65,620.04 | 142,701.76 |
| Long Fibre Composite | 222,615.85 | 6849.72 | 502.48 | 65,620.04 | 295,588.09 |

## 4. Discussion

The purpose of this study was to identify which type of railway sleepers have the lowest emissions and which have the lowest cost over their life cycles. To this end, the study included two sets of life cycle analyses, one focussed on emissions and the other on cost. The impetus for the study is an assessment of the potential of short and long fibre composite sleepers, which have the potential to help reduce plastic pollution by finding an alternative uses for recycled plastic that are long lived and hard wearing. Whether these sleepers are viable, however, also depends on whether they can contribute to meeting net zero goals and whether they are cost competitive with other types of railway sleepers.

The results of this study show that the total embodied emissions over a 100-year period favour short fibre composite sleepers with the lowest overall embodied emissions at 24.85 (or 19.17 with 50% reclamation) tonnes of $CO_2$ equivalent, which is approximately 45% of the emissions of concrete sleepers, the most common sleeper type in use today. In contrast, the long fibre composite sleepers, which are more suited to high-speed rail applications than short fibre composites, had the highest emissions of any sleeper types with emissions more than three times higher than concrete sleepers. These results suggest that there is promise in using short fibre composite sleepers to help reduce the global warming impact of rail infrastructure, but the same is not true for long fibre composite sleepers.

In terms of the calculated emissions for the different types of sleepers, there were a few findings that may challenge conventional thinking. First, it is known that concrete is generally considered to be a large source of $CO_2$ emissions with cement responsible for 8% of global emissions. This would suggest that concrete sleepers might be amongst the worst sleeper types in terms of emissions. However, this study showed that they were in fact the second lowest, and perhaps surprisingly, had much lower total emissions than timber sleepers. The high emissions identified from timber sleepers was another unexpected finding from this study. However, when one considers the energy intensive nature of

the processes that go into creating timber sleepers, the result is more obvious. The high emissions for timber sleepers are largely the result of treating timber to make it a stable material to work with. The process of drying the timber often takes place in kilns that are a major source of $CO_2$ emissions. Kiln drying is the fastest process whereas air drying the timber often costs manufacturers more because air drying takes up a lot of valuable space and more time. Another reason for the high timber emissions is their comparatively short design life and the need to use new fasteners during reinstallation (as it is not possible to achieve a 100% reuse rate in fasteners due to deterioration and damage over time). These factors combine to make timber sleepers one of the less attractive sleeper options from an emissions standpoint.

In terms of the transport emissions component, this study identified that, while all sleepers are roughly the same size, regardless of type, there are significant differences in weight that impact transport costs. Generally speaking, concrete sleepers and long fibre composite sleepers are much heavier than the other sleeper types, so transporting these sleepers to their installation site results in higher emissions than the other, lighter, sleeper types (timber and short fibre composite sleepers). The transportation emissions are proportional to the weight of the sleeper, with the short fibre variant having a weight of approximately 76 kg and the wooden sleeper weighing approximately 65 kg. This equates to emissions of 0.68 tonnes of $CO_2$ equivalents and 1.47 tonnes of $CO_2$ equivalents, respectively (note that timber sleeper emissions are higher owing to the need to replace them more frequently). The long fibre and concrete sleepers are far heavier and their transport emissions are 2.89 tonnes and 7.30 tonnes of $CO_2$ equivalents, respectively.

For the purposes of this study, the installation and removal of sleepers were taken as a single process, because installing a sleeper on an existing track requires the removal of an old sleeper. Although sleepers with a high workability share the same method of installation, the design life of the sleepers ultimately means that the timber sleepers emit 15.25 tonnes of $CO_2$ equivalents, which is significantly higher than that for any of the other sleeper types. The concrete sleepers had the lowest emissions attributed to installation and removal as the method of installation is more specialised and there are less sleepers per 100 m of track, therefore installation is faster and uses less fuel. The emissions associated with the short and long fibre variables are 6.10 tonnes of $CO_2$ equivalent and the emissions associated with installation and removal of concrete sleepers is 5.81 tonnes of $CO_2$ equivalent.

Next, the study analysed life cycle costs to assess whether the more environmentally friendly sleeper options were cost competitive, with industry rarely changing for environmental reasons alone. For example, the shift from timber sleepers to concrete sleepers was largely driven by lower costs [11,26,27]. This price differential was confirmed in this study with, using the design considerations previously described, the total inflation adjusted life cycle cost of timber sleepers is $436,929.72 compared to prestressed concrete, which costs $141,888.44, about one third the cost of the timber they replaced. In addition, although the initial purchase price of individual timber sleepers is already higher than concrete sleepers, the need to replace timber sleepers more often results in an even higher cost. The analyses above also show that there was an unintentional reduction in greenhouse gas emissions at the same time, suggesting that the shift from timber to concrete sleepers was beneficial in several ways [21,26]. However, the global focus on climate change and pollution means that there is now more concern than ever about the impacts of the products and materials we use. So, the question is whether there is a viable replacement product for concrete sleepers that has an even lower environmental impact for a similar economic cost.

This study looked at two such products, short and long fibre composite sleepers. These two products have different uses, with long fibre composites able to replace concrete sleepers for high speed and heavy load tracks and short fibre composite able to replace concrete sleepers for lower speed and load tracks. The emissions analyses suggest that there is no imperative to switch concrete for long fibre composite sleepers, because their emissions are higher than that of concrete sleepers. Moreover, the cost analysis shows

that this sleeper type is also the second most expensive, nearly as expensive as timber sleepers, at a life cycle cost of $427,233.66. Therefore, it seems clear that the long fibre sleeper is neither financially viable nor environmentally desirable and in the absence of future improvements that change this equation, these sleepers are not recommended for extensive use in Australia or elsewhere. However, long fibre composites do have a niche role to play in the market.

This study considered the use of long fibre composites within mainline use and has concluded that they are not viable for that use. However, where they may be used is as bridge transoms that distribute train axle loads directly to bridge girders or where low ballast is a requirement. These specific cases require materials with unique properties, which long fibre composites fulfill better than any other material types considered in this study. Due to the specialised nature of long fibre composites they should not be considered for mainline use but they will likely continue to be used in these special circumstances.

In contrast, short fibre composite sleeper costs in the base case are more like those of concrete, although still somewhat higher at $186,518.12. These results suggest that cost conscious railway industries are likely to shun this product even though they are clearly superior from an environmental standpoint. However, this initial calculation is based on assumptions of no material reuse. Although the short fibre sleeper industry is still in its infancy, and as such there is currently no market for the reuse of these materials, if this product were to become more common, such an industry would clearly develop and short fibre composite sleeper manufacturers suggest that they would be able to reclaim up to 100% of the material for reuse [16]. This study adopted a more conservative approach, assuming that 50% of the material could be reclaimed. Adopting this conservative approach, the analyses found that short fibre composites life cycle costs were reduced 23% to $142,701.76, which is just above the life cycle costs for concrete sleepers (more expensive by $813.32). That is, the short fibre composite sleepers are financially viable and their environmental benefits are substantial, suggesting that a shift to short fibre composite sleepers is warranted anywhere conditions for their use are met (i.e., lower speeds and loads). However, there is likely to be some initial resistance to using these products from the industry, which may resist change due to safety considerations and because the benefits of the resilience and reuse dimensions of these alternative products will not be felt for many years. Research such as this, however, should help to promote the viability of short fibre composite sleepers.

In terms of the breakdown of costs for each sleeper type over its life cycle, transportation costs are relatively high for concrete sleepers, amounting to just over 5% of the life costs for these sleepers and to less than 1% of total costs for all other sleeper types. However, the study did assume that all sleeper types were transported to the site by truck, and if rail transport was used instead, the costs of transport would decrease for all sleeper types. This would obviously impact most on the calculation of total costs for concrete as its starting costs for transport are higher. For fastenings, the highest costs were observed for timber sleepers (8.6% of the total cost of these sleepers), owing to their shorter lifespans, whereas concrete sleepers included no specific fastener cost as their fasteners are included in the purchase price of the sleepers themselves.

This study assumed that the same equipment, machinery and labour would be used for the installation and removal of each sleeper type. Excluded from the calculation of installation/removal costs was the purchase of the equipment or machinery, so the costs reflect the diesel fuel consumed during these processes and are dependant on current diesel fuel prices. The price of diesel fuel in Australia at the time of the analyses was $1.53 per litre. Installation and removal costs were the second largest contributor to life cycle costs for all sleeper types. The cost for installation and removal of a single sleeper was calculated at $121.75, which was the same for all types. Of the $121.75, $110 was from labour expenses for a combination of labourers, machine operators, supervisors, and safety coordinators. The other $11.75 came from fuel consumption. The fuel consumption was calculated from the use of a Caterpillar M315F excavator and Caterpillar 950GC front end loader working for eight hours a day. The M315F excavator was used as it is a wheeled excavator rather

than a tracked excavator and these excavators are often preferred because they can easily be equipped with a hi-rail attachment giving them the ability to drive on train tracks, while the 950GC front end loader is a typical medium sized wheeled machine.

The purchase prices for individual sleepers showed that timber was the cheapest at $90 plus fastenings, followed by concrete at $140 including fastenings, short fibre composite at $203 plus fastenings and the long fibre composite at $650 plus fastenings with the fastening systems costing approximately $20 per sleeper. These prices were obtained via personal communication with representatives from the respective sleeper producers or suppliers. The price of standard gauge short fibre composites is held in commercial confidence so the price of $203 has been estimated using the available price of narrow-gauge sleepers produced from the same material. To extrapolate this price, the six-tenths rule was used with the volume of the two products. Further, while the price of long fibre composites was publicly available it was provided in dollars per cubic metre so this was adjusted based on the assumption that the long fibre composite sleeper would have the same volume as a timber sleeper, due to their similar physical properties. As expected, the price of the individual sleeper was the largest share of the life cycle costs for all sleeper types amounting to 38.65% of the cost of timber, 50.63% of the cost of concrete, 58.66% of the cost of short fibre composite and 82% of the expenditure for the long fibre composite sleepers. Given the importance of purchase price for all sleeper types, any changes to these over time will significantly impact their total costs. Of the sleeper types discussed in this paper, only short fibre composite sleepers have the potential to significantly reduce their costs because this industry is still in its infancy and has yet to benefit from economies of scale, experience curves and competition forces that would likely lead to future cost reductions. Meanwhile, the other sleeper types have been on the market for a long time and their costs are only sensitive to fluctuations in material prices or changes in government legislation or regulation.

Beyond production costs, the conservative parameters used in the analyses above suggest that there are several likely changes in the future that could make short fibre sleepers more cost competitive. If we consider the comparison between concrete and short fibre composite sleeper costs in more detail, for the zero reuse or material/fastener reclamation case, there is a difference in life cycle costs of $44,629.68 in favour of concrete if the sleepers achieve their design life. However, this gap is erased when the 50% reuse case is considered. However, these costs do not include the likely future price reductions that are expected to occur for short fibre composite sleepers, as the total value of any such reductions is unknown. Another possible driver of future price reductions for short fibre composite sleepers would be the broader use of a local (Australian) or global carbon price. As previously discussed, short fibre composite sleepers have significantly lower emissions than their concrete equivalents. A notable carbon price would result in future price reductions relative to concrete sleepers, which would need to pay a higher carbon cost.

Carbon prices have been established in many parts of the world to stimulate a shift to more environmentally friendly alternatives [28]. Carbon prices are a set unit price that liable companies are required to pay per tonne of carbon dioxide that they produce as a result of operations [29]. Australian Carbon Credit Units hold momentary value and can be awarded to companies that undertake projects that either store or avoid carbon dioxide emissions in units of one tonne [30]. Currently the Australian Carbon Credit Unit spot price is $37 [31]. This study has shown that concrete sleepers produce 43.55 tonnes $CO_2$e while short fibre composites produce 24.85 tonnes $CO_2$e, over the same time and for the same length of railway track. This results in a difference of 18.7 tonnes $CO_2$e, meaning that if a rail authority chose to use short fibre composites over concrete sleepers, they may be eligible for Australian Carbon Credit Units to the value of $691.93 per 100 metres of track. Meanwhile, the European Union's price of carbon is $78.06 [32]. If carbon prices within Australia where to increase to this level, rail authorities would be eligible for $1459.78 per 100 m of track (more than the $813.32 higher price for short fibre relative to concrete sleepers). This would make short fibre sleepers more cost effective than concrete sleepers and although the total

financial benefit of just over $600 per 100 m of track may seem trivial relative to the total life cycle costs for each sleeper type, they represent significant costs (or costs savings) over long lengths of track and make short fibre composites clear winners over concrete sleepers financially as well as environmentally.

The combined focus of this study on both embodied emissions and cost implications allows for a more holistic evaluation of the sleeper products. The results clearly highlight the environmental advantages of short fibre plastic composites and based on environmental performance alone this study suggests that these products are the most advantageous. Including a consideration for cost makes the evaluation more challenging. Currently, the reclamation of composite sleeper material is not practiced due to the current product life cycle and the limited available volumes of the product. When considering the life cycle cost for this case, the short fibre composite is outperformed only by the concrete variant. However, calculating the life cycle costs with the assumption of 50% reclamation of materials, the cost advantage of concrete is completely removed. Furthermore, as short fibre composites are completely recyclable it would theoretically be possible for the nominal 50% figure to be surpassed and which would tip the cost advantage towards the short fibre composites.

Although this study provides a relatively comprehensive assessment of the four sleeper types, there were some limitations to the work. These include the heavy reliance on existing literature for the collection of data. Each of the considered studies will have drawn on data that may have had their own limitations in sourcing information. In addition, as composite sleepers are a new product, there is very limited published work available for them. Hence, several assumptions were required about their utility and longevity and only time will tell whether these have been reasonable or not.

## 5. Conclusions

This study sought to determine which types of railways sleepers were the most environmentally friendly and cost effective for use worldwide, but especially in Australia. The results of this study show that short fibre composite sleepers are the clear winners, both environmentally and financially, but only following further industry development and reuse of at least some of the material. Concrete sleepers, which are currently the most common sleeper type used worldwide, have a clear financial advantage over the other sleeper types and perhaps surprisingly, have been one of the more environmentally friendly options available until now. However, the emergence of short fibre composite sleepers, which produce 55% lower emissions over their life cycles than traditional concrete sleepers, have eroded that advantage. Moreover, with the costs of these two sleeper options nearly identical, it is clear that the reduced emissions, coupled with the creation of a new market for recycled plastic that will remove this pollutant from the waste stream (using 54 tonnes of recycled plastic material per km of track installed), results in a clear imperative to shift the dominant form of railway sleepers from concrete to short fibre composite sleepers.

However, the path to this industry transformation is likely to be long and non-linear. First, more example installations need to be rolled out and the performance of the short fibre composite sleepers over time needs to be clearly established. In addition, the boundaries of the conditions in which they can be safely used also needs to be well established. The industry will then need to mature and a reuse market will need to develop. Future work should focus on establishing the viability and safe use of short fibre concrete sleepers and incentivising manufacturers and installers of these sleepers to reward their environmental performance and to encourage a rapid shift from concrete to short fibre composite sleepers wherever the conditions suit their use.

**Author Contributions:** Conceptualization S.R., J.R., S.T. and C.K.; methodology, S.R., J.R., S.T. and C.K.; formal analysis, S.R., J.R., S.T. and C.K.; data curation, S.R., J.R., S.T. and C.K.; writing—original draft preparation, S.R., J.R., S.T., C.K. and M.N.; writing—review and editing, S.R., J.R., S.T., C.K. and M.N.; supervision, S.R. and J.R. All authors have read and agreed to the published version of the manuscript.

**Funding:** This research received no external funding.

**Acknowledgments:** The authors would like to acknowledge the anonymous sustainability rating system professionals who provided feedback on the expert review.

**Conflicts of Interest:** The authors declare no conflict of interest.

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
