# Peer review of "Life Cycle Cost and Assessment of Alternative Railway Sleeper Materials"

_sustainability, doi:10.3390/su14148814_

Round 1
Reviewer 1 Report
The paper analyzes which types of railway sleepers are the most environmentally friendly and cost-effective for use worldwide and with reference to the transport system without Australia.
The introduction clearly presents the investigated problem. The methodology is adequate for the study, compared for all types of sleepers. The procedures are ordered in a logical way.
The method is explained and presented in detail. The results are clearly set out in a logical sequence. The conclusions are supported by the results
Reviewer 2 Report
This manuscript addresses an interesting issue regarding the environmental and cost performance of alternative materials for railway sleeper systems in Australia. A life cycle assessment is conducted for conventional (hardwood and concrete) and alternative (long and short fibre composites) materials focusing on GHG emissions and costs. The manuscript is well-structured and the goal to be addressed is adequately framed. The methodology adopted and the results obtained are clearly presented. Moreover, a discussion section including recommendations is provided. These are the main strengths of the manuscript.
Nonetheless, I include below some comments which I think may improve the overall quality of the submitted document:
1) I understand the authors’ intention in stressing the advantages of short fibre composites. Nevertheless, I think the authors should highlight the different purposes of each alternative (e.g. long fibre sleepers for specific niches, short fibre composites with technical limitations, etc.) from the very beginning of the manuscript to avoid confusion to the readers who may think of four fully interchangeable alternatives;
2) Line 88: I suggest the use of “Life Cycle Assessment” throughout the manuscript, in accordance with ISO standard nomenclature;
3) Line 126: “timber of concrete” should read “timber or concrete”;
4) Line 146/147: “Global warming potential” is not the functional unit of the study… Please revise accordingly;
5) Figure 1: the authors should review the expressions used for the sake of uniformity (e.g. “carbon dioxide equivalents”, “fuel”, “emissions”,…)
6) Lines 162-164: This is not a valid reasoning. The motivation/objectives of the study and its originality should be clearly established within the body of literature available (e.g. wrt to ref [22]);
7) Lines 193-201: I miss the end-of-life stage for concrete sleepers. Do they avoid any environmental burdens e.g. if used as fill material in geotechnical works?
8) The calculations for reclamation scenarios could be better explained, e.g. sleeper figures in Table 5 in comparison to figures in Table 4 (as 60%, 65% and 75% for hardwood, SFC and LFC respectively);
9) Lines 355, 356: this sentence seems to contradict the text in lines 467-469;
10) Line 428: “in Table 5” should read “in Table 6”;
11) Figures 4 and 5: These figures repeat the same information of Tables 6 and 7 and thus could be removed;
12) Line 479: This caption is equal to the caption of the previous table.
Reviewer 3 Report
Manuscript solved interesting view on the environmental relation between rail sleepers and used material for his construction and emission production (CO2). Presented research is on the good quality and I agree with authors conclusions.
I have a few comments about the formal side of the manuscript. Please repair numbering of tables, table no. 6 is in the text twice (line 434 and 479). Maybe authors must check journal tempalte, beacuse in pdf versin main text is narrow opposite the figures and tables.
